# Computational Modeling of Diffusion-Based Delamination for Active Implantable Medical Devices

**DOI:** 10.3390/bioengineering10050625

**Published:** 2023-05-22

**Authors:** Minh-Hai Nguyen, Adrian Onken, Anika Wulff, Katharina Foremny, Patricia Torgau, Helmut Schütte, Sabine Hild, Theodor Doll

**Affiliations:** 1Department of Otolaryngology and Cluster of Excellence “Hearing4all”, Hannover Medical School MHH, 30625 Hannover, Germany; nguyen.minh-hai@mh-hannover.de (M.-H.N.); adrianonken@gmail.com (A.O.); 2Department of Otolaryngology, Hannover Medical School MHH, Carl-Neuberg-Straße 1, 30625 Hannover, Germany; wulff.anika@mh-hannover.de (A.W.); foremny.katharina@mh-hannover.de (K.F.); torgau.patricia@mh-hannover.de (P.T.); 3Department of Engineering, Jade University of Applied Sciences, 26382 Wilhelmshaven, Germany; helmut.schuette@jade-hs.de; 4Institute of Polymer Chemistry, Johannes Kepler University, 4010 Linz, Austria; sabine.hild@jku.at; 5Fraunhofer Institute of Toxicology and Experimental Medicine ITEM, 30625 Hannover, Germany

**Keywords:** active implantable medical devices, cochlear implant, moving boundary diffusion, volume diffusion, interface diffusion, lifetime prediction, COMSOL Multiphysics^®^

## Abstract

Delamination at heterogeneous material interfaces is one of the most prominent failure modes in active implantable medical devices (AIMDs). A well-known example of an AIMD is the cochlear implant (CI). In mechanical engineering, a multitude of testing procedures are known whose data can be used for detailed modeling with respect to digital twins. Detailed, complex models for digital twins are still lacking in bioengineering since body fluid infiltration occurs both into the polymer substrate and along the metal-polymer interfaces. For a newly developed test for an AIMD or CI composed of silicone rubber and metal wiring or electrodes, a mathematical model of these mechanisms is presented. It provides a better understanding of the failure mechanisms in such devices and their validation against real-life data. The implementation utilizes COMSOL Multiphysics^®^, consisting of a volume diffusion part and models for interface diffusion (and delamination). For a set of experimental data, the necessary diffusion coefficient could be derived. A subsequent comparison of experimental and modeling results showed a good qualitative and functional match. The delamination model follows a mechanical approach. The results of the interface diffusion model, which follows a substance transport-based approach, show a very good approximation to the results of previous experiments.

## 1. Introduction

Neural prostheses can restore cognitive, sensory, and motor functions. They stimulate the nervous system through electrical impulses and thus count as active implantable medical devices (AIMDs) [1,2]. The most prominent example of an AIMD is the cochlear implant (CI), which has been implanted 736,900 times worldwide (as of December 2019) [3,4]. Cochlear implants are electronic hearing prostheses that are the most common method worldwide to treat and successfully restore the function of a damaged inner ear or sensorineural hearing loss. Sensorineural hearing loss is caused by the absence or degeneration of hair cells in the cochlea that transmit auditory sensation to the auditory nerve. The CI replaces this function (converting the mechanical stimulus into a neural stimulus) by using an electrode array, which is surgically inserted into the cochlea, to electrically stimulate the auditory nerve so that the individual can perceive a hearing impression [5]. The CI consists of five components: microphone, speech processor, transmitter coil, receiver coil, and electrode array. The microphone picks up the ambient sounds, which the speech processor converts into the appropriate electrical signal. The transmitter coil inductively transmits the electrical signal to the receiver coil, which is located inside the skull. The receiver coil transmits the recorded electrical signal to an electrode array, which is inserted into the cochlea of the inner ear. By targeting the electrodes at different positions in the cochlea, the auditory nerves are electrically stimulated analogous to the stimulation by the hair cells. The electrode array currently consists of up to 24 stimulating electrodes. The electrode material is often platinum (Pt) or a platinum-iridium (Pt-Ir) alloy, which is connected by a platinum-iridium wire [6,7,8,9,10,11]. The wires and a small part of the electrode are coated with silicone, which serves as insulation and therefore does not come into contact with human tissue or the aqueous body fluid perilymph [8,9].

Similar to other AIMDs, CIs can fail. The three main causes of implant failure are leakage, electronic malfunction, and mechanical failure [3,4,12,13]. For example, the receiver coil inside the skull may malfunction and no longer transmit or receive electrical signals. In addition, wires, silicone, or the electrode may develop a crack due to mechanical stress. In most cases, the cochlear implant must be replaced [14,15,16,17,18], which leads to another surgery that results in additional costs for the patients. In order to be able to take statistical precautions, a lifetime prediction is required. Hence, a model is needed that mimics the cochlear implant in a human cochlea. Up until now, there have been few models in the industry, and a lot of research is being conducted to further develop these models [12,13,19,20]. A model that investigates the penetration of fluids into the silicone as well as along the silicone-electrode interface of the cochlear implant is especially lacking.

This study deals with the failure mechanism of electrode array leakage of the cochlear implant, which, according to a study by the Medical School Hannover and the National Institute for Drugs and Medical Devices, is the most frequent cause of failure [3]. The leakage usually occurs at the interface between the electrode and silicone because the perilymph ions can easily diffuse along this interface, which weakens the adhesion forces and thus promotes the detachment of the silicone from the electrode. In addition, the perilymph ions can diffuse into the silicone and change its bulk properties, which leads to hygroscopic swelling. This creates further stresses on the electrode array, leading to mechanical failure. For this purpose, volume diffusion in silicone and interface diffusion along the silicone-electrode interface were studied and modeled. The aim of this work is to establish a general model in which any material can be entered to simulate the delamination process of an AIMD. In this study, the first steps are taken by simulating volume and interface diffusion based on experimental results.

This study is an extension of previous work that experimentally investigated interface diffusion along the copper-silicone intermediate layer [21]. An initial theory for representing diffusion along the interface has been developed. The underlying mathematics refers to boundary conditions according to the Stefan problem solved by Goodman [22,23,24]. In this paper, the modeling approach is extended toward the diffusion-based Stefan problem and implemented as a novel plug-in into the standard multiphysics platform COMSOL [25,26]. For delamination and interface diffusion, two models were created. Modeling delamination follows a mechanical approach, while the interface diffusion model follows a mass transfer-based approach.

For volume diffusion, the COMSOL Multiphysics^®^ database with the diffusion equations according to Fick was used. In order to determine the diffusion coefficient according to Fick, an experimental test was realized. Finally, the simulation results are validated using the experiments.

## 2. Materials and Methods

### 2.1. Volume Diffusion—Potassium Polysulfide Diffusion Test

For the volume diffusion model, the diffusion coefficient of potassium polysulfide ions in polymerized Sylgard 184 was estimated. For this purpose, a copper plate (9 mm × 6 mm, TRU COMPONENTS CFT50/20M 1564016 copper, Conrad Electronic SE, Hirschau, Germany) was prepared and attached to the bottom using Sylgard 184 (0.1 mL; Dow Corning Europe S.A., Seneffe, Belgium) in a cuvette (10 mm × 10 mm × 45 mm; Sarstedt AG & Co. KG, Nümbrecht, Germany). Sylgard 184 was chosen because it is comparable to medical-grade silicone. Sylgard 184 was homogeneously mixed according to the instructions from the prepolymer (1.8 g) and a crosslinker (0.2 g) in a ratio of 10:1 using a SpeedmixerTM (60 s, 1500 rpm; DAC 150.1 FVZ). Possible bubbles are then removed with a cannula. Sylgard 184 was polymerized at room temperature for 48 h. The cuvette was then filled with more Sylgard 184 and also cross-linked so that layers of Sylgard 184 with thicknesses between 1 mm and 10 mm were over the copper plates (Figure 1). Three samples per layer thickness were produced. As a marker for possible swelling of the silicone, the location of the lower point of the formed meniscus was marked [27,28,29,30,31,32,33]. Potassium polysulfide (2.5 mL) was then added to each cuvette. For the preparation of the 2% potassium polysulfide solution, potassium polysulfide (0.4 g, Caesar & Loretz GmbH, Hilden, Germany) was added to deionized water (19.6 g). The solution was mixed for five minutes with a magnetic stirrer (SH-2, serial no. 201411371, Carl Roth GmbH + Co. KG, Karlsruhe, Germany) and filtered. During the experiment, the liquid was changed every 24 h to maintain the desired concentration. The samples were recorded with an HD Webcam C920s Pro (Logitech International S.A., Lausanne, Switzerland) using the software TimeLapse (version 5.4, Microsoft^®^) every five minutes.

In this experiment, copper and the potassium polysulfide solution were chosen because as soon as the potassium polysulfide ions diffused in a sufficient concentration through the Sylgard 184 to the copper plate, the copper began to change color from red to brown, allowing a visualization of the concentration in the Sylgard 184 (Figure 1). Using the copper as an indicator of a certain level of concentration of potassium polysulfide ions in the Sylgard 184, a diffusion coefficient can be estimated from the required time *t* and the thickness of the Sylgard 184 layer over the copper *x* using the Einstein-Smoluchowski relationship (Equation (1)).
(1)$$D=\frac{x^{2}}{2t}$$

### 2.2. Volume Diffusion—Potassium Sulfide Diffusion COMSOL Model

The model for volume diffusion consists of two cylinders lying on top of each other, representing a simplified PDMS drop on a copper substrate, as it is used in the following experiments (Figure 2). The lower cylinder has a thickness of 0.5 mm and a radius of 7.5 mm and represents the copper layer. In direct contact with this layer, a Sylgard 184 cylinder with a radius of 7.5 mm and a varying thickness of 1 mm to 10 mm is defined. The thickness is then increased by 1 mm in each case by means of a parametric sweep, so that several layer thicknesses can be calculated in succession with only one simulation run. The properties of copper and Sylgard 184 used for the volume diffusion model are shown in Table 1.

Two tetrahedral meshes were created for the model. The Sylgard 184 cylinder mesh has a size of 0.6 mm, and the copper layer mesh has a size of 3 mm (Figure 2a,b). For the two cylinders in which diffusion takes place, the physics module “transport of dilute species” with the sub-item “transport property” was selected. For the outer areas of the cylinder, where there is no substance flow, the property “no flow” was added. An infinite source of potassium polysulfide solution was defined at the top surface of the Sylgard 184 cylinder (Figure 2c). The concentration of the potassium polysulfide solution (159.9 $\frac{\mathrm{mol}}{m^{3}}$) was calculated using the amount of substance and the volume of the solution. The starting concentration of potassium polysulfide in Sylgard 184 and the copper layer was set at 0 $\frac{\mathrm{mol}}{m^{3}}$. The criterion for termination was defined as reaching a limit concentration (24.23 $\frac{\mathrm{mol}}{m^{3}}$), at which a visible color change of the copper takes place.

### 2.3. Modeling Diffusion along the Metal-Polymer Interfaces

In the preliminary tests, a Sylgard 184 drop was placed on a copper-coated wafer using a cannula (Figure 3a). The wafer was then immersed in a 2% potassium polysulfide solution. The Sylgard 184 did not have a uniform thickness due to the drop shape. In addition, the Sylgard 184 droplet was not confined by a glass wall as in the above volume diffusion experiment and was in contact with the solution at the side. For this reason, the solution can diffuse along the Sylgard 184-copper interface. During the experiments, two diffusion processes occurred. The interface diffusion started almost immediately, while volume diffusion was observed with a time offset. Both diffusion types occurred in a superimposed manner [21].

The sample geometry (i.e., a droplet with a low contact angle) suggests the following approximation: It can be assumed that the interface diffusion process takes place in the x-y plane, while the volume diffusion takes place primarily in the z-direction.

Due to the slower volume diffusion process starting later, undisturbed interface diffusion was assumed to be the initial phase. The following mathematical interpretation is based on an isolated interface diffusion process.

For interface diffusion, the sample geometry has been approximated as a cylinder with a high radius-to-height ratio. Therefore, interface diffusion, which was initially calculated in 1-D, can be assumed to be isotropic with rotational symmetry and can be used for the poly-dimensional COMSOL Multiphysics^®^ environment.

Mathematically, these processes are described by Fick’s laws. If there is a concentration gradient between at least two adjacent media, mixing occurs until equilibrium is reached [22,34,35]. Temporal and interstationary diffusion processes are described by the second Fick’s law (Equation (2)).
(2)$$\frac{\partial c}{\partial t}=D\frac{\partial^{2}c}{\partial x^{2}}$$

It establishes a relationship between temporal and local differences in concentration c, with the diffusion coefficient D serving as a constant of proportionality. Crank et al. solved this equation for various diffusion systems depending on their boundary conditions [22]. In the case of interface diffusion, the position X of the diffusion front also changes, resulting in a moving boundary problem that exhibits a time-varying boundary condition. These special diffusion processes are Stefan-like problems where two areas are present that are separated from each other. The boundary between the media is variable in time and place. The most prominent examples are melting and solidification processes, with the melting line as the resulting moving boundary between the different phases. [23]. According to Crank and Hill, suitable boundary conditions for such diffusion-based problems are [22]:(3)$$c=0,\;\;t=0,\;\;x>0\;l\frac{\partial c}{\partial t}=D\frac{\partial c}{\partial x},x=0,t\geq 0\;S\frac{dX}{dt}=-D\frac{\partial c}{\partial x},\;\;x=X,\;\;t>0$$

For these conditions, it is assumed that behind the moving boundary, the concentration approaches a stationary distribution at any time, which would occur if the boundary were fixed at that time.

A semi-infinite sheet of a uniform material can be assumed to be in contact with a well-stirred solution with an extent of l from which solutes can diffuse into the sheet. The sheet contains S sites per unit volume, at each of which a diffusing molecule is immediately and irreversibly immobilized. The concentration at any time just below the surface of the sheet is assumed to be that in the solution. The concentration of the immobilized molecules is zero when c=0 and equal to S when c is non-zero.

$X\left(t\right)$ is the value of x at which the concentration c is zero and thus denotes the position of the moving boundary [22,23,24].

A possible solution for such systems was determined by Crank and Goodman using a polynomial approach for the unknown concentration profile c to satisfy the boundary conditions (Equation (4)). The constants a and b are determined from the boundary conditions [22,24].
(4)$$c=a\left(x-X\right)+b{\left(x-X\right)}^{2}\;$$

This solution correlates the position of the boundary according to time, with $X=x\left(t\right)$ describing the position of the moving boundary over time. The faction α includes parameters such as diffusion coefficient, consumption rate, and boundary concentration of potassium polysulfide and was determined empirically in a preliminary test [21].
(5)$$X\left(t\right)=\alpha \surd t$$

This equation forms the basis of the simulation of interface diffusion [24,36]. The α was determined using the experimental results from the preliminary investigation and is 0.027 $\frac{\mathrm{mm}}{\sqrt{\min}}$ (Table 2) [21]. The diffusion coefficient for the interface diffusion was determined in preliminary tests as $D_{i}=$ 1.6761 × 10^−10^ $\frac{m^{2}}{s}$ (Table 2). Both parameters were used for the models.

#### 2.3.1. Interface Delamination Model—Mechanical Approach

The interface delamination model follows a mechanical approach; two rectangles lying on top of each other are defined (Figure 3a,b). The lower rectangle (30 mm × 0.2 mm) symbolizes the copper layer, while the upper left rectangle (14 mm × 1 mm) represents Sylgard 184. The geometry corresponds to a simplified representation of the samples from the preliminary tests by Onken et al., where the Sylgard 184 drop had a diameter of 14 mm and a maximum height of 1 mm (Table 2). The copper layer (200 nm) was sputtered on a 3-inch silicon wafer. In the model, the copper layer thickness was increased and the copper layer width decreased. The properties of copper and Sylgard 184 used for the model are shown in Table 1.

The physics module “solid mechanics” was used in COMSOL Multiphysics^®^. On the bottom side of the Sylgard 184, 23 points with the smallest possible distance of 0.15 µm were inserted, which peel off one after the other due to corrosion of the copper layer. Adhesion is an implicit quantity that is hardly accessible. For this reason, no data could be found in the literature that can be inserted as a parameter in COMSOL Multiphysics^®^. The interface between the Sylgard 184 and the copper is defined as a contact pair. Two tetrahedral meshes were also created, enclosing the Sylgard 184 and the copper layer. Maximum and minimum mesh sizes were defined, with the minimum mesh size near the node points. The transition between maximum and minimum mesh sizes could be defined by the mesh growth rate (Table 3 and Figure 3c).

The equation is summed with the difference between the position of the points on the x-axis and the total length of the Sylgard 184 and passed to a ramp function in order to secure the points detaching one after the other (Figure 3d). If a negative result is passed to the ramp function, no detachment takes place at the point under consideration. As soon as the variable time *t* exceeds a certain timespan, a positive value is passed to the ramp function, resulting in delamination of the account point (Figure 3e). As time progresses, the value of the ramp function becomes larger and larger until unity is reached. This also increases the displacement in the y-direction. For the evaluation, a line integral was added, which sums up all the already detached areas and shows the gap length between the two components along their interface as a function of time. In the future, the ramp function parameters should be replaced by a suitable function that considers the ratio of adhesion and pull-off forces and assesses the grade of delamination, to be passed to COMSOL.

#### 2.3.2. Interface Diffusion Model—Substance Transport-Based Approach

The interface diffusion model follows a substance transport-based approach, altering the mechanical model. No mechanical input from the interface delamination model (mechanical approach) was used for this interface model (substance transport-based approach). An additional rectangle (15 mm × 1 mm) was defined (top right), which represents the potassium polysulfide solution (Figure 4a). Furthermore, an additional intermediate layer (15 mm × 0.01 mm) is defined between copper and Sylgard 184, which has the same properties as Sylgard 184. The properties of copper and Sylgard 184 used for the model are shown in Table 1. Two tetrahedral meshes were also created, enclosing the intermediate layer and the potassium polysulfide solution. As with the delamination, which follows a mechanical approach, the maximum and minimum mesh sizes and the mesh growth rate were also defined for this model (Table 3 and Figure 4b). The mesh size at the boundary between the intermediate layer and the potassium polysulfide solution was set to the minimum. Two physics modules (“deformed geometry” and “transport of diluted species”) are used for the simulation. Under the module “deformed geometry,” the intermediate layer and potassium polysulfide solution are defined as the area to be deformed and their two outer edges as fixed edges. In order to take into account a post-flow of the potassium polysulfide solution into the gap created by delamination, the edge between the potassium polysulfide solution and the intermediate layer is selected under the module “transport of diluted species.” The initial concentration of the potassium polysulfide ions at the interface is 159.9 $\frac{\mathrm{mol}}{m^{3}}$ (Figure 4c).

## 3. Results

### 3.1. Volume Diffusion

In the volume diffusion experiments, no swelling of the Sylgard 184 silicone was observed in any of the samples. In all samples, the height of the silicone meniscus remained unchanged over the entire test period. Therefore, silicone swelling has not been implemented in the simulation. The diffusion distances (1 mm to 10 mm) were predetermined by the differently constructed test specimen. The point in time at which the first color change of the copper became visible was determined (Figure 1). The curve progression reflects an exponential course within the framework of the measurement uncertainty (Figure 5). The results obtained from the volume diffusion tests partly show high scatter per layer thickness. The diffusion coefficient (*D* = 8.901 × 10^−11^ $\frac{m^{2}}{s}$) for the volume diffusion model was estimated using the Einstein-Smoluchowski equation.

The model exhibits a qualitatively similar course as the experiments, but the diffusion progresses faster with the model-determined values than with the experimentally determined values (Figure 5). 

### 3.2. Interface Diffusion

Comparing the simulation result with previous experimental results, it can be seen that detachment at the interface between copper and Sylgard 184 is slower for delamination following a mechanical approach (Figure 6a). For example, the interface should be detached by 0.22 mm after 100 min, according to COMSOL Multiphysics^®^. In the preliminary tests, a distance of 0.24 mm to 0.28 mm was already detached from the copper after 100 min (Figure 6a). In addition, an average coefficient of determination, R^2^ = 0.38, was calculated.

For the interface diffusion model (substance transport-based approach), the average coefficient of determination was R^2^ = 0.95, indicating that the model reproduced well the previous experimental data (Figure 6b).

## 4. Discussion

A comparison of experimental and simulated data shows a clear deviation for volume diffusion. Although the course progression of the simulated data agrees qualitatively with the experimental data, the simulated diffusion runs faster, and the experimental measured values show a high scatter. This is mainly because a diffusion coefficient can only be estimated using Equation (1). Since the copper shows a color change only above a certain limiting concentration, the required time of a diffusing ion is smaller, so the estimated diffusion coefficient is larger. Therefore, the volume diffusion in the experiments is slower, while the simulation runs faster. Additional reasons for this could be explained by differences in measurement accuracy. For example, the layer thickness of the Sylgard 184 layer is subject to a certain degree of inaccuracy due to the manufacturing process. Thus, the layer thickness and meniscus position of the samples vary. A possible improvement of the process could be achieved by using a dispenser that applies a defined volume of silicone. However, the greatest measurement inaccuracy results from the evaluation of the color change of the copper. An interpretation of the color change depends on the light conditions during image recording as well as on personal color perception. Here, a computer-assisted evaluation procedure would be useful. A proper determination of the diffusion coefficient is therefore subject to the greatest measurement inaccuracy and must be improved in the future. Since none of the literature reports values of the diffusion coefficient of potassium polysulfide in Sylgard 184 that are available as a means of comparison, the experimentally determined diffusion coefficient (D = 8.901 × 10^−11^ $\frac{m^{2}}{s}$) is the greatest error influence in the simulation. In most of the literature, the diffusion coefficients of Sylgard 184 were determined for gases [37]. For example, water vapor, hydrogen, and nitrogen have diffusion coefficients of D = 5 × 10^−10^ $\frac{m^{2}}{s}$, D = 1.4 × 10^−8^ $\frac{m^{2}}{s}$, and D = 3.4 × 10^−9^ $\frac{m^{2}}{s}$, respectively [37]. In contrast, there are few studies that discuss the diffusion coefficient of water in Sylgard 184; e.g., Li et al. were able to determine the diffusion coefficient (D = 119.75 × 10^−12^ $\frac{m^{2}}{s}$) of water in Sylgard 184 [38]. An adjustment of the limiting as well as the initial concentration must also be considered. The limiting concentration serves as a termination criterion in the simulation, whereby an incorrect concentration influences the diffusion rate. Since an infinite source is assumed in the simulation, the initial concentration must be constant throughout. Whether this can be guaranteed with a liquid exchange in a 24-h interval must be verified.

A mechanical approach was first chosen to simulate delamination, which is assumed to have a critical influence on the underflow of the liquid into the interlayer. According to the mechanical approach, delamination progresses slower in the model. For this reason, the main causes of delamination are not mechanical stresses, as these occur more slowly than the supposed main cause of delamination.

For interface diffusion, a mass transport-based approach was also designed. The interlayer is corroded by interface diffusion, detached, and removed after reaching a limiting concentration. This results in delamination in the form of a propagating crack. The reason for the delamination is therefore corrosion and the removal of the intermediate layer, which impairs the adhesion of the Sylgard 184 to copper, finally resulting in detachment. The copper corrodes in the potassium polysulfide solution within a few seconds. The corrosion itself can thus be neglected as a time-critical process factor. Accordingly, the most critical factor for delamination is mass transport in the intermediate layer. This transport is well represented by the simulation and coincides with the experimental values from the previous experiments [21].

For the volume model, the Sylgard 184 cylinder mesh has a size of 0.6 mm and the copper layer mesh has a size of 3 mm (Figure 2a,b). For both models describing phenomena in the intermediate layer, the maximum and minimum mesh sizes were defined in Table 3. The transition between maximum and minimum mesh sizes could be defined by the mesh growth rate. No significant differences were found in the results when the mesh size or minimum mesh size decreased. Only the first number after the decimal point has been changed due to the reduction in mesh size. However, if the mesh size increased, the model became less accurate. With a larger mesh size, the volume diffusion model showed a concentration equilibrium at time t = 0, i.e., the potassium polysulfide ion had already reached the copper layer at this time. In the case of interface diffusion, a detachment of the layer was observed at t = 0. With a larger mesh size, volume diffusion, delamination, and interface diffusion in the models progress faster. A significant change in the curve could not be found. With a small mesh size, the simulation becomes more accurate, but the associated computing power and computing time increase. For the volume model, the computing time was not large, as it is a 1-dimensional problem with a fixed boundary. For simulating interface diffusion, the calculation took half a day, so a maximum mesh size and a mesh growth rate were defined. Similar to the minimum mesh size, the maximum mesh size and the mesh growth rate show no significant change from this value. In addition, the calculation time was reduced to two hours.

A future approach would be a more detailed consideration of the cracking in the interlayer in the form of fracture modeling. The current models only consider the simplest case of crack formation by assuming a simple vertical detachment of the silicone from the copper substrate.

In the three models, the intermediate layer was only considered to a limited extent or not at all. The intermediate layer is a composite material formed from Sylgard 184 and copper due to adhesive interactions and has an optimal size of 0.1 nm to 0.5 nm [39,40]. In addition to covalent bonding, polar groups, hydrogen bonding, and secondary valence bonding (dispersion forces) play an important role in the intermediate layer [41,42,43]. Five types (mechanical anchoring, monolayer-monolayer, chemical bonding, diffusion, and pseudodiffusion) are currently known for the polymer-metal interlayer [44]. However, the intermediate layer cannot be clearly assigned in reality because the layer is mainly a combination of the five types [44,45]. In addition, the polymer properties in the interlayer are different from those in the bulk material [46,47,48]. Due to the complexity of the intermediate layer, it was not considered in the volume diffusion model. The model following a mechanical approach was only considered to a limited extent, while in the interface diffusion model, which follows a substance transport-based approach, the intermediate layer had the properties of Sylgard 184. Not considering the intermediate layer in the volume model and delamination model (mechanical approach) can be another reason for the deviation from the experimental results. The interface diffusion model (substance transport-based approach), on the other hand, fits the experimental results well. This may be due to the fact that after reaching a limiting concentration, the interlayer is detached and removed, i.e., the simulation program COMSOL Multiphysics^®^ ignores the properties of the interlayer.

This study was able to show that ions dissolved in water diffuse from the aqueous phase into silicone and, after permeation, react with metallic surfaces and corrode them.

The different permeation possibilities (i.e., diffusion in bulk and diffusion at the interface) were investigated, and a possible model approach was found.

In the future, similar experiments could be carried out with ions contained in body fluids and medically used metals and polymers to ensure better transferability to real implants. Taking into account the delamination mechanisms described in this work, a COMSOL simulation for specific implants can be created with the creation of a database for various medically used materials.

## 5. Conclusions

The aim of this study was to simulate the process of delamination by first considering volume diffusion and interface diffusion individually. Based on experimental results, possible models were designed and tested. For volume diffusion, a qualitatively correct representation of the experimental data was obtained, but the measurement principle and model must be improved in future studies, as currently too high scattering and a slower progression lead to differences between model and experiment. For interface diffusion, the substance transport-based approach could be validated. The mechanical approach describing delamination showed a high deviation from the experimental results, as adhesion forces lacked proper determination. The actual model provides a better understanding of the diffusion-based delamination processes of the PDMS layer from the metal substrate, representing a common failure mechanism of AIMDs. In the future, the focus of further studies on interface diffusion should be on a model based on mass transport. In addition, the intermediate layer and its influence on mass transport should be considered in future simulation models. Since the measurement data from experiment and simulation show that volume and interface diffusion occur superimposed, both types of diffusion must be considered superimposed for a general model since they also correlate with each other. In such a general model, any material can be entered in order to simulate the delamination processes in AIMDs. In order to create this general model, a larger range of materials and their combinations should be analyzed. For this purpose, new experiments are needed. The next tests are currently being planned, which include inert metals (e.g., platinum) that ensure better transferability to real implants.

## Figures and Tables

**Figure 1 bioengineering-10-00625-f001:**
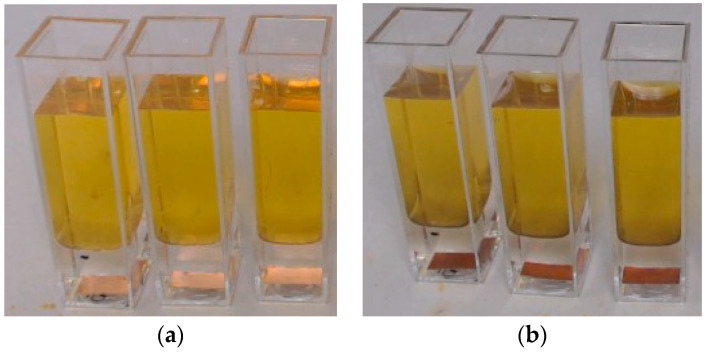
Three samples with a diffusion distance of 10 mm were used to determine the concentration of potassium polysulfide ions in Sylgard 184, as indicated by the color change (**a**) at the start of the experiment and (**b**) at the end of the experiment.

**Figure 2 bioengineering-10-00625-f002:**
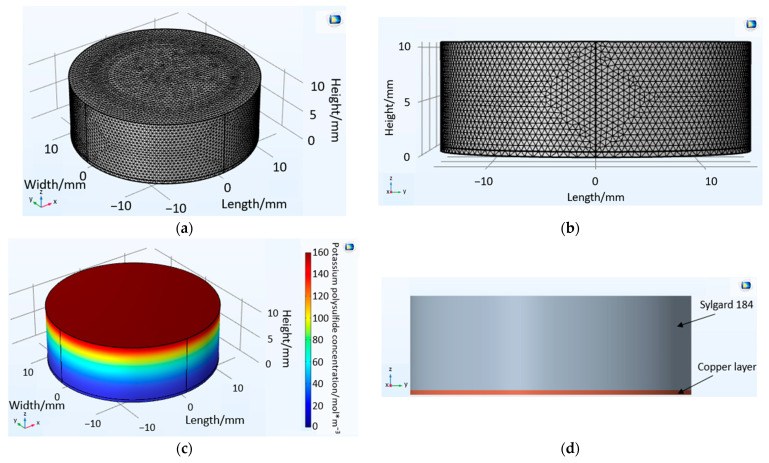
Two superimposed cylinders of the volume diffusion model showing the tetrahedral mesh (**a**) in 3D views and (**b**) in the side view. In addition, the volume diffusion model (**c**) with the potassium polysulfide concentration and the (**d**) side view clearly show the boundary between Sylgard 184 and the copper layer.

**Figure 3 bioengineering-10-00625-f003:**
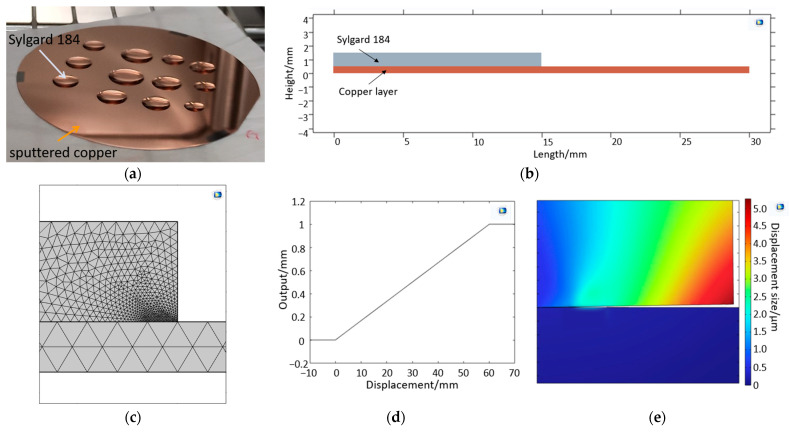
(**a**) The Sylgard 184 and copper layer from the preliminary tests [21], (**b**) the geometry of the Sylgard 184 and copper layer for the mechanical delamination model, (**c**) the tetrahedral mesh at the detaching points, (**d**) the ramp function, and (**e**) the delamination of Sylgard 184 from the copper layer.

**Figure 4 bioengineering-10-00625-f004:**
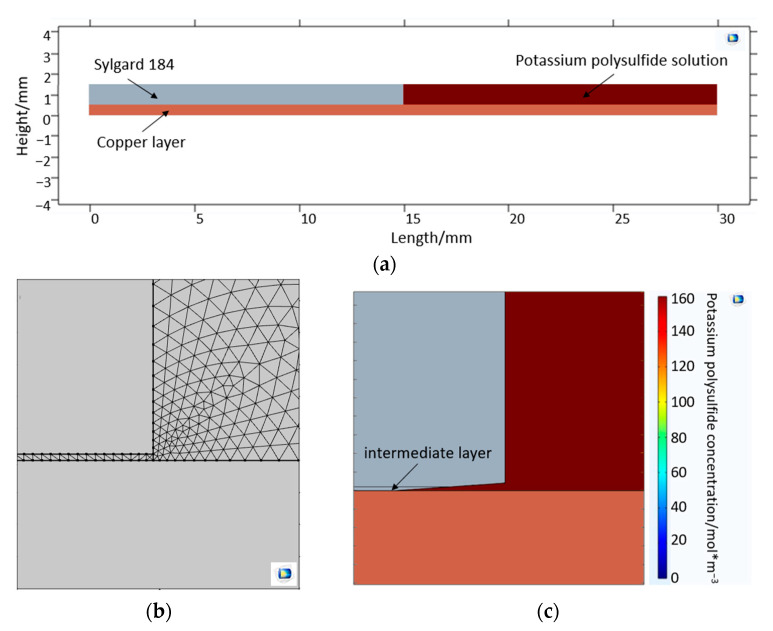
(**a**) The geometry of Sylgard 184, the copper layer, the intermediate layer, and the potassium polysulfide solution; (**b**) the tetrahedral mesh at the boundary between the intermediate layer and the potassium polysulfide solution; and (**c**) the interlayer corroded by the interface diffusion, which is detached and removed.

**Figure 5 bioengineering-10-00625-f005:**
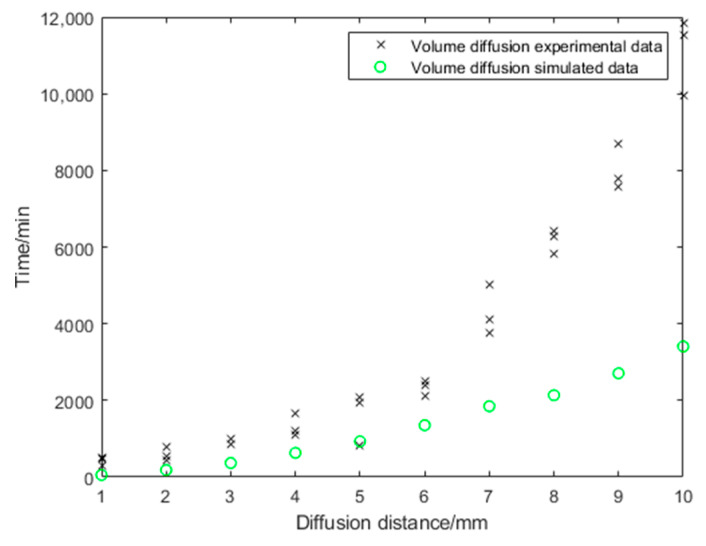
Volume diffusion: comparison of experimental and simulated data.

**Figure 6 bioengineering-10-00625-f006:**
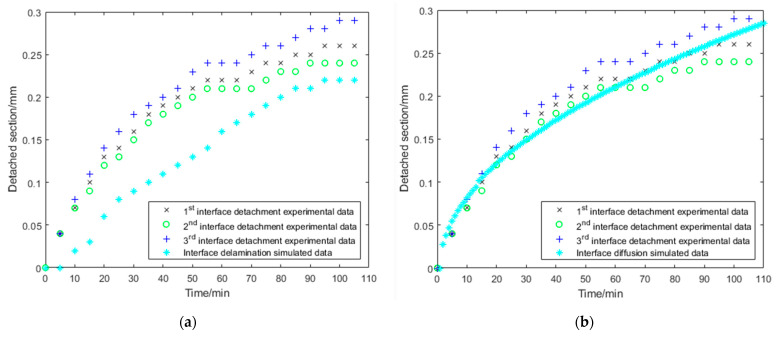
Mechanical delamination versus interface diffusion: experimental data in comparison to the (**a**) interface delamination model and (**b**) interface diffusion model.

**Table 1 bioengineering-10-00625-t001:** Material properties used in the COMSOL simulation.

Properties	Copper	Sylgard-184
Density	8960 [kg/m^3^]	2329 [kg/m^3^]
Poission ratio	0.35	0.28
Youngs Modulus	110 × 10^9^ [Pa]	170 × 10^9^ [Pa]

**Table 2 bioengineering-10-00625-t002:** Experimentally determined coefficients for the COMSOL simulation [21].

Name	Value	Description
α	0.027 mm/$\sqrt{\min}$	Boundary coefficient
*r*	0.014 m	Radius of the PDMS droplet
$D_{i}$	1.6761 × 10^−10^ m^2^/s	Interface diffusion coefficient

**Table 3 bioengineering-10-00625-t003:** Mesh properties of the COMSOL simulation.

Mesh Parameter	Sylgard 184/Intermediate Layer Mesh	Copper Layer/Potassium Polysulfide Solution Mesh
Minimum mesh size	6 × 10^−3^ mm	0.54 mm
Maximum mesh size	0.06 mm	0.3 mm
Maximum mesh growth rate	1.3	1.5

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
