# Peer review of "Computational Modeling of Diffusion-Based Delamination for Active Implantable Medical Devices"

_bioengineering, 2023, doi:10.3390/bioengineering10050625_

Round 1
Reviewer 1 Report (Previous Reviewer 2)
The authors sufficiently addressed most of my concerns. The only remaining reservation is the potentially misleading usage of "active medical implants". Since, as the authors acknowledged, no voltage or current was involved, the model was not "active". The authors should add their response to the manuscript and clarify what they studied was in fact a dummy for a typical AIMD material profile.
Author Response
Dear Reviewer 1,
thank you so much for your enduring patience with us and our article. Enclose we are sending you our response to the reviewers comments
thank you in advance for your time
Best regards
Minh-Hai Nguyen

Reviewer 2 Report (New Reviewer)
It is an interesting study on the active implantable medical devices. Work is well-written, easy to follow and clear to understand. I do not have any concerns related to this study Only some slip of pens to be corrected. As a result, I suggest acceptance of this work
In this study, simulations of the delamination process have been presented.
Topic is relevant to the field and is of importance in biomedical devices, particularly when considering implants. Additionally, it addresses some issues that are currently under investigations. Unfortunately, this study does not provide a complete understanding of the delamination process. As to the originality, the present work is extension of the previous studies of the same authors. Nevertheless, it is worth accepting this work because a lot of computations was realized and results were clearly explained (previous studies provide just preliminary data).
As mentioned above, present study performed more detailed investigations that previous ones and obtained results were clearly explained (previous studies provide just preliminary data).
The further improvements and needs are also discussed in the manuscript in section discussion and conclusions.
Conclusions are in agreement with findings of this study.
Figures and tables are understandable and readable.
Only some minor errors must be corrected.
Author Response
Dear Reviewer 2,
thank you so much for your enduring patience with us and our article. Enclose we are sending you our response to the reviewers comments
thank you in advance for your time
Best regards
Minh-Hai Nguyen

Reviewer 3 Report (New Reviewer)
General comments
The overall aim of the study was to produce and test models simulating the delamination process in active implantable medical devices for volume diffusion in silicone and interface diffusion along the silicone-electrode interface. Deviation between experimental data and simulated data was observed for volume diffusion, although qualitative agreement was observed for the course progression. For the interface diffusion model, the substance transport-based approach was validated.
General comments
In general, the paper is sufficiently well written and clear. The study is timely, given the focus on addressing CI failure. The findings are novel (although this could be clarified – please see comment below), the methods appear sound, and the Tables and Figures are on the whole presented clearly. The manuscript will be of interest to readers of Bioengineering. I have no major concerns, and only some minor comments that the authors may wish to consider.
Specific comments
Line 93: “This study is a continuation of the work of Onken et. al, which experimentally investigated interface diffusion along the copper silicone intermediate layer [19]. In these studies, a theory for the representation of the diffusion in the interface could be found.”
Given that the current work is an extension of previous work, in order to clarify the novelty of the current study please provide more detail regarding how the current work is novel as it is a continuation of the work of Onken et al.
Conclusions section and end of abstract: What are the implications of the findings for active implantable (note: there is a typo “implantatable” in the abstract) medical devices, specifically for cochlear implants? For example, how exactly can the current findings be used to help prevent implant failure?
Figure 2 panel c: The text of the y axis title is rather small. I would suggest making it larger.
Author Response
Dear Reviewer 3,
thank you so much for your enduring patience with us and our article. Enclose we are sending you our response to the reviewers comments
thank you in advance for your time
Best regards
Minh-Hai Nguyen

Reviewer 4 Report (New Reviewer)
In this work, the authors reported a continuation of the published work (Ref. 19). The basis of the simulation program was the finite element method (FEM) and couple physics-based models were created. The substance transport-based approach was partially validated. The tested shapes were simple. However, the improvements of the optimized method in this work show limited innovation. The simulations carried out were routine. The associated amount of work is very minimal. It seems like a tinkering study without significant adjustments to the experiment design or a new conceptual or scientific insight. The improvements are required in text style and graphs. Some suggestions are as below.
(1) The originality is under concern. The manuscript has too much background description. The authors need to clearly point out the innovation and performance superiority of the work relative to similar devices in the article. In addition, what is the scientific significance of the work?
(2) Another major flaw is that the manuscript is full of typos, mistakes in both grammar and syntax, and in many cases incomprehensible. The authors should carefully edit the manuscript to avoid sloppiness, especially tense.
(3) There is nothing regarding “active implantable medical devices”. The demonstration needs more quantitated details.
(4) What are the advantages and disadvantages over other devices with different structures? The necessary description need to be provided.
(5) The authors claimed that “This study is a continuation of the work of Onken et. al, which experimentally investigated interface diffusion along the copper silicone intermediate layer [19].” As a whole, very limited extended exploration on theory novelty or functionality was considered.
(6) The quality of manuscript doesn't meet the merits for publishing in this journal. It might fit more to some specified journal. I would recommend to reject or transfer.
In this work, the authors reported a continuation of the published work (Ref. 19). The basis of the simulation program was the finite element method (FEM) and couple physics-based models were created. The substance transport-based approach was partially validated. The tested shapes were simple. However, the improvements of the optimized method in this work show limited innovation. The simulations carried out were routine. The associated amount of work is very minimal. It seems like a tinkering study without significant adjustments to the experiment design or a new conceptual or scientific insight. The improvements are required in text style and graphs. Some suggestions are as below.
(1) The originality is under concern. The manuscript has too much background description. The authors need to clearly point out the innovation and performance superiority of the work relative to similar devices in the article. In addition, what is the scientific significance of the work?
(2) Another major flaw is that the manuscript is full of typos, mistakes in both grammar and syntax, and in many cases incomprehensible. The authors should carefully edit the manuscript to avoid sloppiness, especially tense.
(3) There is nothing regarding “active implantable medical devices”. The demonstration needs more quantitated details.
(4) What are the advantages and disadvantages over other devices with different structures? The necessary description need to be provided.
(5) The authors claimed that “This study is a continuation of the work of Onken et. al, which experimentally investigated interface diffusion along the copper silicone intermediate layer [19].” As a whole, very limited extended exploration on theory novelty or functionality was considered.
(6) The quality of manuscript doesn't meet the merits for publishing in this journal. It might fit more to some specified journal. I would recommend to reject or transfer.
Author Response
Dear Reviewer 4,
thank you so much for your enduring patience with us and our article. Enclose we are sending you our response to the reviewers comments
thank you in advance for your time
Best regards
Minh-Hai Nguyen

Round 2
Reviewer 4 Report (New Reviewer)
Thank the authors for the response. For each comment, the authors responded so much little and did not reply them clearly and directly. There is no intuitive explanation of the key to the problem. Compared to very similar work, the improvements of the optimized method in this work show limited innovation. The simulations carried out were routine. The associated amount of work is very minimal. It is a tinkering study without significant adjustments to the experiment design or a new conceptual or scientific insight. I don't think the innovation of this work is enough to make this work published in this journal. The work cannot bring broad readership.
The originality is under concern. The manuscript is full of typos, mistakes in both grammar and syntax, and in many cases incomprehensible. Such as “The faction α includes parameters like diffusion coefficient, consumption rate and boundary concentration of potassium polysulfide and was determined empirically in preliminary test.” and “For the evaluation, a line integral was added, which sums up all the already detached areas and shows the gap length between the two components along their interface as a function of time.”, there are too many like them. In the Materials and Methods and the Results and Discussion, the description of past experimental results requires the use of the past tense. There is nothing regarding “active implantable medical devices”. More tests are needed for the demonstration.
As a whole, very limited extended exploration on theory novelty or functionality was considered.
The manuscript is full of typos, mistakes in both grammar and syntax, and in many cases incomprehensible. Such as “The faction α includes parameters like diffusion coefficient, consumption rate and boundary concentration of potassium polysulfide and was determined empirically in preliminary test.” and “For the evaluation, a line integral was added, which sums up all the already detached areas and shows the gap length between the two components along their interface as a function of time.”, there are too many like them.
This manuscript is a resubmission of an earlier submission. The following is a list of the peer review reports and author responses from that submission.
Round 1
Reviewer 1 Report
Good revision and interesting work.
Author Response
Dear reviewer,
thank you very much for your feedback. We were pleased with the positive feedback
Yours sincerely
Minh-Hai Nguyen

Reviewer 2 Report
The authors experimentally measured the diffusion coefficient of potassium polysulfide in Sylgard 184 silicone and presented two computational models of a silicone droplet delamination experiment from prior work. The models are generalizable for studying the delamination of implantable device housing caused by correlated detachment and diffusion. Albeit the result and the methodology are relevant to the research community, the following issues should be addressed before the manuscript can be considered for publication:
1. The appropriate title of the paper should be “Computational modeling…” rather than “Mathematical…”, since the mathematical models (Fick’s diffusion, Stefan problem etc.) were adopted from prior works, and the original work in this manuscript was to implement the models in COMSOL, a computational software. In fact, Equation (4), the general solution to the mathematical problem, was given incorrectly (should be “… + b(x-s)^2”). The mistake was carried down from Ref. 19 and indicated the manuscript did not involve close examination of the mathematical models.
There was no interplay between volume diffusion and delamination in this paper (line 202-204), and the title should not imply so by “delamination-based volume diffusion”.
Also, why “active” implantable devices? The modeling does not involve any feature of the device being active.
2. In the abstract, the authors claimed “… testing procedures … are lacking in bioengineering, as body fluid ingress happens both into the polymeric substrate and along the metal polymer interfaces.” This is simply not true. Both FDA and EMA have extremely detailed guidance on the durability of implantable devices, of which metal and polymer are two most common housing materials. There are also countless labs and companies around the world specializing in the design and implementation of testing procedures for medical implants.
3. On page 2 the authors seemed to have used “silicone” and “silicon” interchangeably, whereas they presumably meant silicone. Bare silicon is rarely, if ever, exposed to the biological environment. If body fluid is in contact with silicon, the implantable device has already failed.
4. Section 2.1 missed one critical link – at what concentration of K2Sx did the copper exhibit a color change? The authors need to keep in mind that Equation 1 does not imply a binary boundary of K2Sx concentration, and the concentration at the copper surface was never zero once the solution had been filled in. Therefore, the threshold concentration of color change should be a premise of the measurement, not a result of modeling (line 170). The time point t = x^2/2D was merely the time when the concentration at the copper surface reached 1-erfc(1)≈0.16 of the solution, which was an arbitrary value.
5. The authors should clarify why the volumetric diffusion experiment and modeling are relevant to implantable devices. One-dimensional Fick’s diffusion with a fixed boundary is trivial. Is Sylgard 184 comparable to medical grade silicones? Is K2Sx surrogate of any ingressive factor in vivo? Or was the experiment just for finding the diffusion coefficient of K2Sx in silicone for the droplet experiment?
6. “the diffusion progresses faster with the model… ” (line 325). “the simulated diffusion runs faster and the experimental measured values show a high scatter” (line 349). However, Figure 5 indicated the opposite – model was slower.
7. The information was poorly organized throughout the manuscript. Several examples:
The description of the “first interface diffusion model” was incoherent. Section 2.3 described the dynamics of diffusion with a moving boundary, but there was no K2Sx solution in Figure 3, so what was diffusing? Later in the manuscript, it was referred to as “the mechanical approach” (Figure 6 caption and Discussion), the volumetric model was referred to as Model 1, and the first model was referred to as Model 2 (line 407)… The authors should clearly state: (1) what physical mechanism(s) caused the delamination in the “first model”, and (2) what was the physical meaning of the ramp function. The authors should also name the three models in a consistent and intuitive manner.
The narrative of Section 2.3 was also confusing. It was not immediately clear to the reviewer that the two interfacial models were studying another geometry from a previous work. The authors should include an illustration of the geometry so the paper would be self-sufficient.
Table 2 and Table 3 were not mentioned in the main text. Some properties were reported in these two tables, but the domains were not specified. Moreover, the diffusion coefficient reported in Table 3 was not the value measured by the authors (line 323).
Some sentences read obscure: Line 230-232; Line 263-265; Axis label of Figure 3(c)
8. Other miscellanea:
Equation (3) second line right hand side – should be partial derivative of x.
Equation (3) – please define l, S and X right after the equation. An example is not a definition. Equation (4) – variable b here and the b in Table 3 are presumably different variables?
Equation (5) – the description of alpha is very vague. Please properly define alpha and s.
The authors should be consistent in their use of “interface diffusion” and “interfacial diffusion”.
Citations [22] and [25] are too broad.
Line 15: The first “AIMD” should be parenthesized.
Author Response
Dear reviewer,
Thank you for your feedback. We have changed our manuscript accordingly. Enclosed we send you the point-by-point response
your sincerely
Minh-Hai Nguyen

Reviewer 3 Report
The manuscript entitled “Mathematic modeling of delamination-based volume diffusion and interfacial diffusion for active implantable medical devices”. The authors have reported the theory of interface diffusion along the copper silicone intermediate layer by using COMSOL Multiphysics. The authors should submit the final vision of the manuscript instance of revision one. Therefore, I recommend reviewing this paper after submitting the final version.
Author Response
Dear reviewer,
thank you for your feedback
your sincerely
Minh-Hai Nguyen

Reviewer 4 Report
The manuscript describes a COMSOL model to study the delamination in active implantable medical devices. Delamination is presented as one of the most critical failure mechanisms in soft medical devices. The work is a continuation of Reference 19, an experimental study on interface diffusion. Some of the model parameters are obtained by preliminary experiments. The second model that was developed resulted in good agreement with the experimental data. Although the analysis results are not comprehensive, the manuscript could be useful for researchers in this field. Before publication, it must include analysis to provide more insight into the delamination mechanisms and the factors playing a role in delamination. My major revisions are as follows; 1) In the first model, a mechanical approach was used to model the delamination. However, the boundary conditions, forces acting on the model, and the adhesion between the layers are not defined. Provide a more detailed description of the COMSOL model. 2) In model two, where mass transport is taken into account, are the mechanical inputs included from model one, or is corrosion applied alone without the mechanical inputs? 3) Title says this is a "mathematical model" which suggests an analytical model is developed but it should be more accurately titled as a "computational model" 4) The importance of the mesh size selection on the model accuracy should be quantified. 5) In the discussion how the findings of this study can be generalized and applied to other scenarios must be included. 6) The detachment results should be analyzed for a larger range of material combinations. The factors playing a role in delamination should be properly quantified. As a minor feedback, the manuscript, especially the table captions should be edited for English as they include some grammar mistakes.My recommendation is major revisions with new experiments needed.
Author Response

(The authors gave the same response as above.)
